# Hormesis and Oxidative Distress: Pathophysiology of Reactive Oxygen Species and the Open Question of Antioxidant Modulation and Supplementation

**DOI:** 10.3390/antiox11081613

**Published:** 2022-08-19

**Authors:** Mariapaola Nitti, Barbara Marengo, Anna Lisa Furfaro, Maria Adelaide Pronzato, Umberto Maria Marinari, Cinzia Domenicotti, Nicola Traverso

**Affiliations:** Department of Experimental Medicine, University of Genoa, Via L.B. Alberti 2, 16132 Genova, Italy

**Keywords:** oxidative (di)stress, hormesis, ROS, antioxidant supplementation/modulation, GSH, NRF2, aging, inflammation, cancer, degenerative diseases

## Abstract

Alterations of redox homeostasis leads to a condition of resilience known as hormesis that is due to the activation of redox-sensitive pathways stimulating cell proliferation, growth, differentiation, and angiogenesis. Instead, supraphysiological production of reactive oxygen species (ROS) exceeds antioxidant defence and leads to oxidative distress. This condition induces damage to biomolecules and is responsible or co-responsible for the onset of several chronic pathologies. Thus, a dietary antioxidant supplementation has been proposed in order to prevent aging, cardiovascular and degenerative diseases as well as carcinogenesis. However, this approach has failed to demonstrate efficacy, often leading to harmful side effects, in particular in patients affected by cancer. In this latter case, an approach based on endogenous antioxidant depletion, leading to ROS overproduction, has shown an interesting potential for enhancing susceptibility of patients to anticancer therapies. Therefore, a deep investigation of molecular pathways involved in redox balance is crucial in order to identify new molecular targets useful for the development of more effective therapeutic approaches. The review herein provides an overview of the pathophysiological role of ROS and focuses the attention on positive and negative aspects of antioxidant modulation with the intent to find new insights for a successful clinical application.

## 1. Reactive Oxygen Species

Free radicals are chemical species characterised by the presence of at least one unpaired electron on their outer orbitals. Among them, some species are free radicals but not ions (e.g., hydroxyl radical HO^•^), some species are ions but not free radicals (e.g., hydroxyl ion, OH^−^), others are both free radicals and ions (e.g., superoxide anion O_2_^•^^−^) [1]. Free radicals are characterised by reactivity towards other chemicals, which leads to their biological activity and depends on their half-life [2].

The reactivity of free radicals is in any case due to the unpaired electron, which usually gives the radicals a strong oxidant power, and sometimes a reductive power [3]. The above-mentioned free radicals are oxygen-centred, i.e., the unpaired electron is on the oxygen atom of the chemical species, and they are considered as the main free radicals acting in the biological environment. From their interconversion reactions hydrogen peroxide (H_2_O_2_) can be produced. H_2_O_2_ is not a free radical but has similar reactivity and plays a fundamental role in the pathophysiology of many diseases. Both oxygen-centred radicals and hydrogen peroxide are referred to as reactive oxygen species (ROS) [4].

Mitochondrial activity crucially contributes to ROS generation. Indeed, during mitochondrial respiration a proportion of oxygen in the respiratory chain is partially reduced, by only one electron, to O_2_^•^^−^ [5]. Even though this was once considered an “imperfection” of the mitochondrial respiration, today it is known that cells are able to adapt by modulating the amount of oxygen free radicals in order to regulate many essential physiological processes [6]. ROS are also generated during the biotransformation of xenobiotics, mainly in the redox reactions carried out by CYP450 (Cytochrome P450) system [7,8].

Importantly, cells possess several enzymes able to produce ROS. The most typical enzyme that produces free radicals is NOX (nicotinamide adenine dinucleotide phosphate, NADPH oxidase), whose main product is O_2_^•^^−^ [9]. This enzyme was firstly discovered in phagocytes that are capable of generating a high amount of ROS in order to complete substrate digestion with no damage to themselves or to the tissues, in a process known as “oxidative burst”, carried out by the isoform NOX2 [10]. NOX2 is localised on the plasma membrane and it is internalised when the phagocytic vacuole is formed, so that O_2_^•^^−^ production is confined in intracellular vesicles [11]. Many NOX isoforms were gradually discovered in a wide variety of cell types, where they modulate several different cell functions by regulating the intracellular levels of O_2_^•^^−^ and H_2_O_2_ [12,13]. In the central nervous system, NOX-derived ROS have been implicated in the regulation of cell survival/death, migration, differentiation, proliferation, synaptic plasticity and neuroinflammation [12].

Moreover, some enzymes, such as xanthine oxidoreductase (XOR) do not produce ROS in their usual form but become able to produce them in particular conditions. Indeed, xanthine dehydrogenase (XDH) utilises NADP^+^ as a cofactor to which the electrons are released, while its variant xanthine oxidase (XO) is unable to use NADP^+^, and uses molecular oxygen as an electron acceptor leading to O_2_^•^^−^ formation [14]. The transition from XDH to XO occurs under ischemia condition, so that it is widely believed that this conversion and the subsequent abundant production of O_2_^•^^−^ is the main participant in the damage triggered by ischemia-reperfusion [15].

ROS can also derive from non-enzymatic reactions. For instance, haemoglobin can be oxidised to methemoglobin leading to the generation of O_2_^•^^−^ [16]. Additionally, many sugars and lipids can undergo spontaneous autoxidation generating O_2_^•^^−^ in the presence of oxygen and transition metal ions (especially copper and iron ions), the latter acting generally as redox catalysers [17,18].

As an exogenous source, the ionising radiation causes the homolytic cleavage of H_2_O leading to the generation of ROS which oxidise macromolecules, impairing cells and tissues [19].

## 2. Protection against ROS Generation

Several mechanisms of protection have been developed to counteract the toxic effects of ROS [20].

Among the enzymatic defences, one of the most important is superoxide dismutase (SOD) [21]. Three isoforms of SOD are known: copper-zinc SOD (CuZnSOD), which is present mainly in the cytoplasm; manganese SOD (MnSOD), located in the mitochondria; and extracellular SOD. They catalyse the same dismutation reaction of O_2_^•^^−^ into H_2_O_2_ and molecular oxygen. It has been demonstrated that mice lacking MnSOD die perinatally with a cardiomyopathy due to a massive ROS generation [22].

In addition, catalase (CAT) converts H_2_O_2_ to O_2_ and H_2_O, and peroxiredoxins (Prx) reduce alkyl hydroperoxides and H_2_O_2_ to their corresponding alcohol or H_2_O [23]. Moreover, thioredoxins (TRX) protect cells from oxidative stress, reacting with ROS and reducing oxidised proteins. They also serve as hydrogen donors to the TRX-dependent peroxide reductases. TRX1 is expressed in the cytoplasm and the nucleus while TRX2 is expressed in the mitochondria [24].

Among antioxidant molecules, the tripeptide glutathione (GSH), formed by glutamic acid, cysteine, and glycine, plays a pivotal role in ROS detoxification [25]. The reduced form of glutathione (GSH) reaches millimolar concentrations in the intracellular compartment of all types of cells while the oxidised form (GSSG) is estimated to be less than 1% of the total GSH [26]. The majority of GSH is detected in the cytosol (90%), while mitochondria contain nearly 10% and the endoplasmic reticulum have a very small amount [27,28]. By means of GSH peroxidase, (GPX) activity peroxides are reduced and GSH is oxidised to GSSG [29], playing a crucial role in maintaining the intracellular redox balance and thiol status of proteins [30]. Afterwards, glutathione reductase regenerates reduced glutathione with the oxidation of NADPH to NAD^+^ [31]. Thus, the availability of NADPH is crucial; the major source of NADPH in cells is the pentose phosphate pathway (PPP), and in particular Glucose 6 phosphate dehydrogenase, whose genetic X-linked deficiency is the cause of a spectrum of human pathological conditions [32,33]. Since GSH/GSSH is the main redox couple inside the cells, the measurement of this ratio represents an index of the global intracellular redox potential [34]. In addition, glutathionylation of proteins [35] regulates enzyme activity, transport activity, signal transduction and gene expression through redox-modulated nuclear transcription factors such as Activator Protein 1 (AP-1), Nuclear Factor-kappaB (NF-kB) and p53 [36] whose DNA-binding involves critical cysteine (Cys) residues that need to be maintained in a reduced form [37]. GSH also takes part to the detoxification reactions in reactions catalysed by isoforms of GSH-*S*-transferase (GSTs) [25,38,39].

GSH can be released by cells and proteolysed in the γ-glutamyl cycle [40]. The fragments of GSH generated by γ-glutamyl-transpeptidase (γGT) are probably the form of interchange of GSH between organs [41]. The liver is probably the major source of GSH for other organs. Since the GSH secreted outside the cells is rapidly degraded by γGT, GSH levels in extracellular fluids are virtually absent.

Thus, plasma antioxidant power can be attributed mainly to two molecules: bilirubin and uric acid. Bilirubin is generated from biliverdin by biliverdin reductase, a reaction that seems not to have other physiological functions than bilirubin generation, thus providing plasma of this important antioxidant [42,43]. Notably, the production of bilirubin in the intracellular compartment accounts for protective antioxidant activity that plays an important role in counteracting endothelial dysfunction-based pathologies [44]. Uric acid accumulates in the plasma of primates at a concentration well above that of many other mammals due to the inactivation of the enzyme uricase in primates [45]. Unfortunately, both bilirubin and uric acid also possess toxic effects, but their antioxidant power overwhelm the danger that they can represent.

Other well-known antioxidant molecules are some vitamins, such as vitamin A, *_L_*-ascorbic acid (vitamin C), α-tocopherol (vitamin E) [46], polyphenols and minerals such as selenium and zinc [47]. Vitamin C and vitamin E are the most studied and important dietary antioxidants. Vitamin C is a water-soluble vitamin that is fundamental for the biosynthesis of collagen and many other biomolecules [48]. It is an important antioxidant in the blood and is indispensable for vitamin E regeneration [49]. Vitamin E is a lipid-soluble antioxidant localised in the plasma membrane and it is able to protect cell membranes against lipid peroxidation [50]. The importance of the antioxidant role of vitamin E is proved by many studies conducted over a century from its discovery [51].

Among polyphenols, resveratrol, obtained from grape skins and red wine, has been classified as antioxidant, cyclooxygenase inhibitor, peroxisome proliferator-activated receptor stimulator and endothelial nitric oxide synthase (NOS) inducer [52].

## 3. Modulation of Redox Equilibrium

Intracellular redox potential is a homeostatic parameter of the cells, namely the balance between oxidant species and antioxidant defences, and the ratio GSSG/GSH is one of the most reliable markers of the redox equilibrium in a biological context, as above reported [53]. The imbalance between ROS production and antioxidant ability leads to a condition of oxidative stress, a term firstly coined by Helmut Sies [54]. This condition leads to cell and tissue damage and has been implicated in many pathological conditions, from aging, to diabetes, atherosclerosis, many degenerative diseases and cancer [55,56,57,58,59,60]. The role of oxidative stress in each of these pathological conditions can be different: permissive, causative or complementary, and this distinction is often a matter of debate. However, the majority of scientists agree that oxidative stress is at least a contributing mechanism in each of the mentioned diseases, as explained later in Section 4.

### 3.1. Redox Signaling

Physiological ROS can act as second messengers able to induce cell adaptative responses by activating multiple signal transduction pathways. H_2_O_2_ is recognised as the major player in redox regulation of biological activities [61] and like calcium [62] is a versatile pleiotropic physiological agent. The intracellular production of H_2_O_2_ is stimulated by growth factors, chemokines or physical stressors while its removal is achieved by efficient reducing systems [63]. The maintenance of nanomolar concentrations of H_2_O_2_ leads to reversible oxidation of thiolate groups in target proteins [64] which, by altering protein activity, localization and interactions, contributes to modulate cell proliferation, differentiation, migration and angiogenesis [65]. This steady-state of low-level H_2_O_2_ and the related physiological redox signalling is called “oxidative eustress” [66].

Redox signalling starts with the reaction of H_2_O_2_ with Cys residues of protein targets to form the sulfenate (SO^−^) that can subsequently lead to intramolecular or intermolecular disulfide (SS) formation or glutathionylation (SSG) of the reactive cysteines [67]. When O_2_^•^^−^ is generated concomitantly with NO, peroxynitrite (ONOO^−^) is formed leading to nitration of tyrosine (Tyr) residues in proteins, causing functional changes. NO *per se* can lead to *S*-nitrosylation by interacting with Cys yielding protein persulfides or polysulfides [68]. The formation of disulfides and of glutathionylated, nitrosylated and persulfidated Cys is reversibly regulated by the TRX system or by the GSH system [69].

Redox signalling can also occur via reversible methionine oxidation [70] and lipid oxidation [71]. In addition, other molecules, such as non-coding RNAs or microRNAs, have been shown to contribute to redox signalling since they are down or up-regulated by ROS and because they can modulate enzymes involved in ROS generation and disposal [72].

Thus, physiological levels of ROS can modulate cell survival, proliferation, differentiation and death activating a wide number of protein kinases such as protein kinase C (PKC) isoforms, mitogen-activated protein (MAP) kinase, MAPK, extracellular signal-regulated kinase (ERK)1/2, phosphoinositide 3-kinase/serine-threonine kinase (P13K/Akt), protein kinase B (PKB), protein tyrosine phosphatases (PTPs) and several transcription factors e.g., NFkB, Hypoxia-Inducible Factor (HIF), p53 and Nuclear Factor Erythroid 2 p45-Related Factor 2 (NRF2) [73,74].

Notably, it has been demonstrated that moderate levels of the oxidant species lead to cell adaptation to stress or resilience, a phenomenon known as hormesis [75].

While physiological levels of H_2_O_2_ are important for cell signalling, supraphysiological concentrations (above 100 nM) cause unspecific oxidation of proteins and irreversible damage to macromolecules, such as lipids and DNA triggering growth arrest and cell death. In this context, as described by Dianzani [76], products derived by cell injury can serve as secondary signalling molecules such as 4-hydroxynonenal and other reactive aldehydes that are generated during lipid peroxidation and are able to react with proteins. Elevated levels of ROS mediate DNA damage that has been widely characterised in mutagenesis and cancer development [77]. Moreover, it has been demonstrated that ROS can induce global DNA hypomethylation, promoter hypermethylation and histone modification favouring carcinogenesis [78]. The above-described condition is associated with pathological conditions and can be recognised as “oxidative distress” (Figure 1).

### 3.2. The Nuclear Factor Erythroid 2 p45-Related Factor 2 (NRF2)

NRF2, discovered in 1994 [79] is a key regulator of cellular redox state. It plays a fundamental role in the antioxidant cell response regulating the inducible cellular defence system by the expression of more than 250 genes [80].

In resting condition, the Kelch-like ECH-associated protein 1 (Keap1) provides the constant ubiquitination of NRF2 and its proteasomal degradation, necessary to maintain NRF2 at low basal level. NRF2 turnover is rapid, about 15–30 min, and provide basal expression of NRF2 target genes in normal conditions [81]. Oxidative or electrophilic stressors are responsible for Keap1 conformational changes that allows NRF2 to move from the cytosol into the nucleus [82]. In the nucleus NRF2 dimerises with small MAF protein (sMAF), binds to the antioxidant (ARE) or to the electrophile responsive (EpRE) elements and activates the transcription of its target genes among which NAD(P)H quinone oxidoreductase 1 (NQO1), heme oxygenase 1 (HO-1), enzymes involved in GSH synthesis (glutamate cysteine ligase catalytic, GCLC, and modifier, GCLM, subunit), SOD and CAT as well as genes involved in xenobiotic metabolism and drug transport [83].

Other mechanisms of NRF2 regulation has been described such as the redox independent proteasomal degradation mediated by Glycogen Synthase Kinase-3β (GSK-3β) and β-Transducin Repeat-Containing Protein (β-TrCP) [84], and the p62 (SQSTM1)- dependent stabilization [83]. NRF2 can be also regulated at transcriptional level due to the presence in its promoter region of regulatory sequences such as Xenobiotic Responsive Element (XRE) and XRE/ARE-like sequences which enable NRF2 autoregulation [85,86]. Moreover, it has also been demonstrated that oncogenic mutation of Kirsten Rat Sarcoma viral oncogene homolog (KRAS), v-raf murine sarcoma viral oncogene homolog B1 (BRAF) and c-myelocytomatosis oncogene (c-Myc) transcriptionally regulate NRF2 at the mRNA level [87]. Furthermore, epigenetic modification in the promoter region as well as microRNAs (miRNAs), long non-coding RNAs (lncRNAs), DNA and histone modifications contribute to changes in the regulation of NRF2 and of its target genes [88,89], as recently reviewed [90]. Notably, the binding of NRF2 to ARE sequences can be regulated in order to modulate the transcription of target genes. Indeed, ARE motif is recognised not only by NRF2 but also by other regulator proteins such as BTB and CNC homology 1 (Bach1) and c-Myc which ca be modulated as well to regulate NRF2 activity [91,92,93].

In healthy cells, through its transcriptional network, NRF2 is an essential defence mechanism to maintain cellular homeostasis and redox metabolism, protecting against oxidative and electrophilic stressors, preventing cancer initiation, and regulating the inflammatory process. Notably, NRF2 activation also through an integrated cross talk with other transcription factors drives hormetic cell responses crucial in the survival of biological systems [94].

On the other hand, the loss of function of NRF2-dependent pathway can play a pivotal role in many acute and chronic pathological conditions, in neurodegenerative (Alzheimer disease, AD [95] and Parkinson disease, PD [96]), cardiovascular diseases [97], in ischemia/reperfusion injury [98], in metabolic disorders (i.e., diabetes) [99] as well as in aging process [100]. Therefore, NRF2 activation in healthy cells and in pathological conditions related to oxidative stress and inflammation have been proposed in order to prevent or attenuate the disease, as described later in Section 5.3.

### 3.3. MicroRNAs

MicroRNAs (miRNAs) are short non-coding RNA molecules produced by eukaryotic cells and may act as mediators of cell signalling. Among them, some miRNAs, named “redoximiRs”, are involved in the maintenance of intracellular redox homeostasis [101]. In fact, if, on one hand, miRNAs are sensitive to oxidative stress [72,102], on the other, they can target the gene expression of several ROS generators, of redox sensors and antioxidants [103,104]. In addition, it has been demonstrated that miRNA biogenesis is modulated by ROS and, in turn, miRNA biogenesis is able to produce ROS [105].

Therefore, ROS and miRNAs are engaged in a vicious circle where they are strictly connected and mutually influenced. For instance, it has been shown that miR-320 overexpression stimulates proliferation and inhibits apoptosis of ischemic cerebral neurons by reducing ROS production. This effect is due to miR-320-mediated NOX2 inhibition and the concomitant increase of SOD, CAT and GPX [106]. Additionally, miR-652 has been found to inhibit NOX and its overexpression prevents apoptosis of brain tissues of rats with middle cerebral artery occlusion [107]. Moreover, the gene expression of many antioxidant enzymes is targeted by miRNAs. As an example, miR-17-3p inhibits SOD2 and TRX increasing damage of human retinal pigment epithelial cells [108]. It has been found that miR-21-5p, miR-23a-3p and miR-222-3p target SOD2 in cardiomyocytes and have been suggested as new circulating biomarkers of heart failure [109]. Additionally, miR-509 targets SOD2 and its downregulation in breast cancer is associated with tumour progression [110]. On the contrary, miR-21 overexpression induced tumourigenesis via SOD3 and Tumor Necrosis Factor (TNF)-α inhibition [111]. Other miRNAs such as miR-433 can reduce GSH synthesis by targeting γ-GCL and promoting Transforming Growth Factor (TGF)-ß-dependent fibrogenesis [112]. The dysregulation of several redoximiRs affecting the expression and/or activity of molecules of antioxidant pathways (Sirtuin 1, Forkhead box proteins, Keap1/NRF2) or effector enzymes (e.g., GPX-1, SOD1/2, HO-1) or ROS producers (e.g., NOX4) have been found to be involved in the pathogenesis of the metabolic syndrome by modulating lipid uptake and storage, adipogenesis, cholesterol metabolism, inflammation, mitochondrial activity, glucose homeostasis, and endothelial function [113].

Notably, little is known about the role of miRNAs as modulators of redox homeostasis at the level of the whole organism. In this context, it is important to consider that miRNAs are secreted from all types of eukaryotic cells, packaged into vesicles and released to the extracellular medium where they may reach distal tissues to modulate gene expression and influence cell-to-cell communication [101].

## 4. Oxidative (di)Stress and Disease

Oxidative stress or, as recently reported, oxidative distress is associated with the development and/or maintenance of a variety of pathological conditions.

With regard to the etiopathogenesis, ROS-related diseases can be divided into two categories: those in which oxidative stress is the primary cause, or those in which oxidative stress contributes to the evolution of a disease induced by other factors [114].

The first group includes diseases in which ROS, originating as a result of exposure to radiation or chemicals [115] or oxidised lipids [116], induce an inflammatory response that, in turn, further modifies cell signalling and stimulates cytokine production inducing a fibrotic response.

The second one includes pathologies in which oxidative stress is consequent to inflammatory responses induced by other causes.

As a result of their involvement in metabolism and detoxification, the liver and kidneys are the main targets of ROS attack: in the liver, ROS induce membrane damage of hepatocytes leading to cirrhosis and in the kidney, ROS stimulate the secretion of pro-inflammatory cytokines causing nephrosis [117]. ROS can also act as neurotoxic agents by reducing neuron excitability and are responsible for cardiac myopathy via mitochondrial damage [117].

Therefore, a causal or co-causal role of ROS has been demonstrated in several diseases and in particular those that are age-related. In fact, aging itself, being associated with the major production of ROS, is the period of life characterised by the highest incidence of redox-related diseases [118,119]. For instance, it has been demonstrated that the onset of oxidative stress is strictly related to the incidence of atherosclerosis, hypertension, diabetes, many neurodegenerative diseases and cancer [114].

### 4.1. Oxidative Stress and Aging

In 1956, the Harman’s “Theory of aging” postulated that molecule and cell damage due to free radical attack is responsible for aging process, recognising also a major contribution from mitochondrial ROS leaking [120]. However, plenty of literature has been published over the years both supporting and against Harman theory, as revised by Vina and colleagues [121]. The controversy comes especially from the evidence that the degree of oxidative stress and ROS generation have been found increased with aging in some experimental models [122]. Importantly, in 2010 Ristow and coworkers demonstrated the importance of a mild oxidative stress in promoting metabolic health favoring longevity through the induction of molecular mechanisms of defence, thus highlighting the concept of mitochondrial hormesis [123]. Indeed, mice deficient in MnSOD showed increased incidence of different pathologies but not a decrease in life span [124]. Thus, in a more complex vision, the conditions of oxidative (di)stress, can be related to aging, while moderate levels of ROS associated with stress adaptation and enhanced cell survival, can be responsible for an increased life span [125,126,127].

Notably, among the oxidative reactions involved in aging it is important to consider the role played by the Maillard reaction, namely the chemical reaction between reducing sugars and proteins leading to the generation of advanced glycation end-products (AGEs) [128]. This reaction is crucially favored by oxidative stress and in turn favors cell damage, in a vicious circle that is pivotal in aging and age-related diseases [129,130,131]. For instance, AGE accumulation is involved in cardiovascular diseases (CVD) [132,133], and neurodegeneration [134]. Moreover, in the context of successful aging, the importance of maintaining antioxidant hormetic mechanisms has been investigated [100].

### 4.2. Oxidative Stress in Inflammation and Inflammatory Diseases

ROS generation plays a pivotal role in the accomplishment of inflammation, being essential in completing the phagocytosis of pathogens, in favouring tissue repair, and in modulating different biological functions in immune cells. However, the dysregulation of ROS production and the onset of oxidative stress generates and favours tissue damage associated with inflammation-related diseases. The dichotomy “good ROS/bad ROS” in inflammatory responses is well described by Barry Halliwell [135].

Phagocytes are able to generate a high amount of ROS [9] thanks to the activity of NOX whose role in inflammation has been recently reviewed [136]. The loss of NOX2 activity due to the genetic loss of function of one of the NOX2 subunits leads to chronic granulomatous disease (CGD) [137]. Patients show a deficit in killing pathogens, due to the impaired respiratory burst, and are unable to respond to fungal and bacterial infection [138].

In addition, it has been well demonstrated that NOX1 and NOX2 are crucially involved in monocyte to macrophage differentiation and in the gain of M2 and tumour-associated macrophages (TAM) phenotype [139]. Moreover, we also demonstrated that other molecular mechanisms related to mild conditions of oxidative stress such as oxidation of phospholipids or uremic toxin stimulation contribute to M2 polarization [140,141].

Importantly, NOX-derived ROS also plays an important role in the regulation of antigen presentation and interferon production, and this aspect has been recently reviewed [136].

Furthermore, a link between NOX activity, especially NOX1 and NOX2, and the activation of inflammasome, in particular on Nucleotide-Binding Domain (NOD)-like receptor protein 3 (NLRP3) and caspase 1, is a matter of debate. On one hand, there is evidence of a role of NOX2, as inhibitor of NLRP3, which is able to limit IL-1β production. The observation of the occurrence of hyper-inflammatory manifestations such as lupus-like manifestations in patients with CGD [142] supports this evidence, and points out a possible role of NOX2 as immunomodulatory [143]. On the other hand, other authors report a pro-inflammatory activity of NOX2-derived ROS due to the activation of inflammasome NLRP3 [144]. Similar observations have been provided for NOX4 that activates NLRP3 playing a prominent role in the onset and progression of inflammatory bowel disease [145]. Interestingly, an indirect mechanism involving the modulation of fatty acid oxidation has been demonstrated in NOX4-dependent NLRP3 activation [146]. The existence of opposite results highlights the complexity of inflammasome regulation, and the role played by oxidative stress deserves further investigations.

Notably, a new mechanism of tissue regeneration involving NOX2 and resolving inflammation has been recently reported. Indeed, NOX2 released from macrophages in extracellular vesicles can be internalised by damaged neurons by endocytosis and favours axonal regeneration through the activation of PI3K/Akt kinases [147].

Among the possible other sources of ROS in phagocytes, mitochondria play an important role. Even though ROS derived from mitochondrial metabolic activity can be related to functional responses in macrophages, such as Toll-like Receptor (TLR)4 [148] or TLR2 [149] dependent activation, often ROS generation from mitochondria correlates with uncontrolled hyper inflammatory conditions and tissue damage, and its inhibition seems to be protective against immune-mediated damage. Indeed, it has been reported that mitochondria-derived ROS are involved in the overproduction of cytokines in response to Lipopolysaccharide (LPS) in an auto-inflammatory disorder due to TNFα Receptor mutations [150]. Moreover, ROS overproduction results in the inflammation of synovial and bone erosion and the reduction of mitochondrial activity can play a protective and therapeutic role [151]. Moreover, recent evidence demonstrates that mitophagy can reduce NLRP3 by reducing ROS generation from dysfunctional mitochondria with a crucial role played by early endosomal compartment [152].

Interestingly, H_2_O_2_ produced by monoamine oxidase (MAO), located on the outer mitochondrial membrane and characterised for their impact on neuronal cells, has also been reported to modulate NRLP3 in macrophage through NF-kB activation, and its pharmacological inhibition was proved effective in endotoxemia treatment [153].

#### ROS and Hyperinflammation in Coronavirus Disease 2019 (COVID-19)

A hyper inflammatory state, often referred to as cytokine storm, is associated with the severe cases of severe acute respiratory syndrome coronavirus 2 (SARS-CoV-2) infection. In these cases, from the activation of alveolar macrophages in response to the virus, neutrophils are recruited and undergo a strong NOX2 activation and ROS production that leads to lung damage [154]. Moreover, oxidative stress works as a driving force in favouring neutrophil extracellular trap (NET) formation, thus reducing T cell mediated antiviral response [155]. In addition, the hyper activation of neutrophils and the improper activation of NOX2 has been related to capillary leaks and microvascular thrombosis in critically COVID-19 patients.

However, other authors hypothesised a crucial role of mitochondria in the onset of severe COVID-19. Indeed, the increased plasma level of ferritin associated with the infection favours intracellular oxidative stress through Fenton reaction and mitochondria dysfunction, contributing to both endothelial dysfunction and platelet activation and increasing the risk of capillary leaking and thrombosis [156].

### 4.3. Oxidative Stress and Metabolic Diseases

The development of metabolic diseases reached epidemic proportion in the last decade and the role played by oxidative stress both in the onset of metabolic syndrome and in the development of steatohepatitis, obesity, diabetes and vascular complications is widely recognised [157]. Different mechanisms have been demonstrated to contribute to oxidative stress in metabolic syndrome, such as NOX activation s, glycoxidative stress, high-density lipoprotein (HDL) dysregulation and alterations of antioxidant systems. For instance, both proliferation and differentiation of adipocytes involve ROS production [158] that has been proved to be due to NOX4 activation [159] and mitochondrial dysregulation [160]. The excessive production of ROS in obesity leads to proinflammatory activation of macrophages with production of TNF-alpha, IL1 and IL6 [161], and glycoxidative stress can also play a role [162]. Then, ROS overproduction and inflammation favour dysfunctions of pancreatic beta (β) cells on one hand [163] and contribute to insulin resistance on the other [164], resulting in type II diabetes [165].

Notably, the development of non-alcoholic steatohepatitis (NASH) is strictly dependent on the unbalance of redox state. A recent work suggests NASH to be considered as a highly complex syndrome in which environmental factors as well as genetic hallmarks are interrelated, but that is inevitably centred on mitochondrial and endoplasmic stress and ROS overproduction [166]. Additionally, oxidative stress is responsible for producing oxidative stress-derived epitopes that have been recognised as key activators of innate immune response, and drive chronic hepatic inflammation and the associated liver fibrosis and/or cirrhosis [167]. Of note, the modulation of NRF2-dependent cell adaptation has been proposed to counteract NASH progression, as described later in Section 5.3.1. In addition, redox unbalance as well as inflammation associated with metabolic syndrome favour the onset of cardiovascular complications often associated with metabolic diseases [157].

### 4.4. Oxidative Stress and Degenerative Diseases

As already mentioned, oxidative stress has a pivotal role in the pathogenesis of different degenerative diseases, including cardiovascular diseases, metabolic disorders or neurodegenerative diseases.

The oxidation-dependent activation of specific molecular pathways leading to cell damage has been pointed out, from seminal observation on the role of mitochondria alteration in diabetic complications [168] to more recent analysis of the role played by oxidative stress in micro- and macro-angiopathy [169]. Thus, the role of mitochondria-derived ROS is fundamental in CVD [170], and mitochondria homeostasis has been proposed as a therapeutic target also in the context of aging and age-associated diseases, as recently reviewed [171]. Nonetheless, the dysregulation of NOX isoform activity, which plays a fundamental control of vascular functions (oxygen sensing and vascular tone regulation), is involved in CVD [172], in atherosclerosis [173], and in the development of kidney disease associated with hypertension and diabetes [174].

Notably, the role of crosstalk among NOXs and other sites of ROS production in the pathogenesis of vascular diseases has been recently widely reviewed [175].

The increase in ROS generation, especially mitochondria-derived ROS, is also pivotal in neurodegenerative diseases as proved for AD [176], PD [177] and Huntington diseases [178].

However, the role of NOXs both in physiology and in pathology has been well demonstrated also in the nervous system. Indeed, regional NOX isoform activation with specific regional localization has been related to acute and chronic brain disease [179]. In agreement, we also demonstrated that the specific Receptor for AGE (RAGE)- and PKC-dependent activation of NOX2 is responsible for retinoic acid-induced differentiation in neuron-like cells [180,181], but the same pathway activation leads to cell damage in presence of glycated products [182,183,184].

On the other hand, the role played by the loss of hormesis and the impairment of antioxidant defences needs to be considered in the pathogenesis of degenerative diseases. The lack of function of NRF2 and its dependent genes has been widely highlighted, in atherosclerosis [185] or neurodegeneration. In this context, we also demonstrated the involvement of NRF2/HO-1 axis in endothelial survival against hyperglycaemic insult [44] and also contributed to point out the crucial role played by the reduction of GSH in liver damage [186].

Importantly, the upregulation of NRF2 negative regulator such as Bach1 is related to the onset and development of degenerative pathologies [187,188,189]. In this context, we demonstrated that the impairment of Bach1-dependent HO-1 activation increases sensitivity of neuronal like cells towards oxidative stress [190].

### 4.5. Oxidative Stress and Cancer

Oxidative stress is involved in all stages of cancer development [114]. In addition, it is the main mechanism through which both radio and chemotherapy exert their cytotoxic action [191].

During the earlier phases of carcinogenesis, ROS stimulate the activation of stress-response pathways and induce both genetic mutations and epigenetic alterations [192]. In fact, the oxidation of guanine to 8-oxoguanine (8-oxoG) is the major form of DNA damage, and therefore, it is considered a biomarker of oxidative DNA damage [193,194]. Moreover, oxidative stress can also induce DNA damage by modulating the activity of enzymes involved in chromatin remodelling, inducing histone modifications [195] and by altering miRNA biogenesis and sequence [196,197,198].

During tumour growth, ROS can play a key role in promoting the metastatic potential of primary tumours [192]. In fact, it has been reported that ROS stimulate the expression of metalloproteases (MMPs), enzymes that favour tumour invasion [199] via p38MAPK activation [200].

At the later stages of cancer progression, tumour cells acquire an increased ability to counteract oxidative injury [201] and support cell survival.

To this aim, cancer cells have evolved several strategies such as: (i) increasing the expression of antioxidant enzymes (e.g., MnSOD, TRX and HO-1) [80,202,203,204] and (ii) reprogramming their metabolism [205]. However, in several tumours a constitutive activation of NRF2 pathway leading to the up-regulation of antioxidant genes, metabolic enzymes and membrane associated-transporters has been documented [206]. Moreover, the constitutive activation of NRF2 is crucial for tumour survival since many tumours have been found to display high levels of ROS as a result of the activity of oncogenes such as K-Ras, c-Myc and Mesenchimal epithelial transition factor (c-Met), Epidermal Growth Factor Receptor (EGFR), Platelet-derived growth factor receptor (PDGFR) or Src [207,208,209] via NOX activation [210].

Therefore, the presence of antioxidants sustains tumour progression but also helps cancer cells to resist to ROS-mediated cytotoxic action of therapies [205,211,212].

In order to sustain uncontrolled growth, tumours take advantage of aerobic glycolysis, known as the “Warburg effect,” to produce ATP. This is a dynamic and reversible process through which the most malignant cells, can adopt oxidative phosphorylation (OXPHOS) depending on the environmental conditions [213]. In fact, cancer cells are strongly engaged in mitochondrial metabolism [214,215] and the use of OXPHOS inhibitors has been demonstrated to counteract tumour growth [216]. In addition, recently it has been demonstrated that this metabolic shift is strongly involved in the acquisition of both chemio- and radio-resistance [205,217] and that the viability of therapy-resistant cancer stem cells is due to their ability to maintain an efficient OXPHOS metabolism [218,219]. Recently, our study proposed that PKC-α inhibition might be a useful strategy to sensitise neuroblastoma stem cells to chemotherapy by inducing a metabolic switch from OXPHOS to aerobic glycolysis, decreasing GSH levels and stimulating ferroptotic death [218].

#### GSH and GSH-Related Pathways in Cancer Progression

As above reported, cancer cells become able to survive under pro-oxidation conditions by up-regulating GSH content and GSH-related pathways [212].

Several studies have reported that neoplastic cells displaying high metabolic rate and ROS over-production are characterised by a marked increase in γGT activity that, favouring intracellular cysteine supply, sustains GSH resynthesis [220,221]. In fact, the low but constant γGT-dependent production of H_2_O_2_ increases DNA instability and can stimulate cancer cell to proliferate and survive [221]. Moreover, γGT-expressing neoplastic cells, due to their high levels of GSH, are able to counteract the pro-oxidant effects of traditional drugs.

With regard to GSH biosynthesis, many tumours show an enhanced expression of GCLC and GCLM. Indeed, the expression/activity of GCLC and/or GCLM is high in patients with lung cancer, squamous cell carcinoma and renal cell carcinoma [222,223,224] and can be associated with chemoresistance condition [222,225].

Furthermore, it has been demonstrated that cancer cells are able to up-regulate the expression of GPXs whose five members are selenoproteins (GPXs 1–4 and GPX6) and are crucially involved in the defence against oxidative stress. In particular, it has been found that GPX1 and GPX2 modulate proinflammatory mediator synthesis and release, and their up-regulation is critically involved in the development and promotion of colorectal cancer and [226]. The only extracellular isoenzyme GPX3 has still a controversial role since it can act as both as tumour suppressor and pro-survival protein [227]. Indeed, on one hand, low levels of GPX3 have been detected in plasma and/or tissue samples from non-small-cell lung cancer (NSCLC), glioblastoma, hepatocellular carcinoma and colorectal carcinoma associated with a high rate of lipid peroxidation and poor patient outcome [228]. On the other hand, GPX3 expression is increased in clear cell adenocarcinoma, ovarian cancer and leukaemia [227].

The selenoperoxidase GPX4 was firstly isolated by Ursini and coll. [229] and it has been defined the “cornerstone” of the defence against lipid peroxidation [230]. In fact, GPX4 missing or insufficient activity triggers ferroptosis [231,232,233].

Recent studies reported that tumour suppressor protein p53 sensitises cells to ferroptosis [234] by down-regulating Solute Carrier Family 7 member 11 (SLC7A11), the light chain subunit of the Xc− transporter that mediates the uptake of extracellular cystine in exchange for glutamate [235].

Moreover, p53 increases the transcriptional activity of glutaminase 2 (Gls2), a mitochondrial glutaminase catalysing the hydrolysis of glutamine to glutamate [236]. On the other hand, p53 when up-regulates CDKN1A/p21 (cyclin dependent kinase inhibitor 1 A) expression was found to delay the induction of ferroptosis by preserving adequate intracellular GSH levels and maintaining GPX4 activity [237].

## 5. Antioxidants Used for Therapy

As above reported, oxidative (di)stress is crucially associated with the onset of several human diseases both as a primary and a secondary cause. Therefore, in order to successfully counteract the redox related pathologies many in vitro studies and clinical trials have been carried out to test the impact of antioxidants including natural and artificial ROS scavenging molecules.

Many antioxidants have been investigated for therapeutic approaches and some of them are currently undergoing clinical trials. These include SOD and GPX mimics, NOX inhibitors, GSH precursors, NRF2 modulators, and dietary antioxidants [114].

### 5.1. SOD Mimics, GPX Mimics and NOX Inhibitors

Several SOD mimics have been developed and well described [238,239]. Although their specific mechanism of action is to remove superoxide anion, most are not specific and can also reduce other reactive oxygen or nitrogen species such as ONOO^−^, peroxyl radical and H_2_O_2_.

With regard to Mn porphyrins, well-known SOD mimics, the phase I trial is ongoing for the use of MnTDE-2-ImP5+ for the treatment of amyotrophic lateral sclerosis, and no toxicity has been observed [114]. Moreover, a recent phase I clinical trial of SOD mimetic GC4419 for the treatment of oral mucositis showed a protective and safe effect in patients with oropharyngeal cancer treated with radio and chemotherapy [240]. GC4419 is a highly stable Mn (II)-containing macrocyclic complex that is able to remove specifically O_2_^−^^•^ without reacting with other ROS.

Among GPX mimics, the seleno-organic compound ebselen is best known [241]. This mimetic shows a specificity for H_2_O_2_ and membrane-bound phospholipid and cholesterol hydroperoxides. Clinical trials with ebselen are ongoing in Meniere disease (phase III, NCT04677972) [242] and those with ALT-2074 in diabetes and coronary heart disease (phase II, NCT00491543). Notably, ebselen has been reported to inhibit also NOX activity [243].

NOX inhibitors include peptidic and small-molecule inhibitors that inhibit NOX enzyme activity or suppress the assembly of the NOX2 enzyme [244]. Among small molecules, diphenyleneiodonium (DPI) is the first identified and commonly used NOX inhibitor even if is not selective since it might target different flavin-dependent enzymes such as XO and NOS. Although the development of NOX inhibitors is highly promising, as highlighted by Schmidt et al. [245,246], it remains challenging to identify compounds that target NOX in a specific manner. Indeed, more research is needed to develop NOX inhibitors for the treatment of oxidative stress-related disorders.

### 5.2. Modulation of GSH Levels

#### 5.2.1. Strategies to Maintain GSH Homeostasis

Strategies finalised to maintain or replenish intracellular GSH levels are based on the employment of agents that provide cysteine, the limiting amino acid in GSH biosynthesis, or GSH esters.

N-acetylcysteine (NAC) is found in plants of the Allium species, especially in the onion, and it is the most studied antioxidant agent. In animal studies and clinical trials, NAC is being investigated for prevention or treatment of several diseases. Even if NAC, especially as oral administration, is considered a safe compound, the results obtained from these studies are sometimes contrasting or incomplete [247].

Notably, like other antioxidants, NAC supplementation might be harmful in the case of cancer patients undergoing chemotherapy because it can increase cancer cell survival by enhancing GSH levels [248]. In this regard, NAC administration to a mouse model of malignant BRAF-mutated melanoma increased the number of lymph node metastases and enhanced GSH amount in the metastatic sites without altering the number and size of primary tumours [249].

In addition, ester derivatives of GSH have been synthesised and evaluated for their efficiency. Although the results obtained from in vivo and in vitro studies [250,251] demonstrate that the supplementation with GSH esters is safe and efficient to increase GSH levels, no clinical trials have been reported.

#### 5.2.2. Decreasing GSH Levels to Increase Sensitivity of Cancer Cells to Therapy

Based on the evidence that cancer cell survival is facilitated by high amounts of GSH [212], several approaches aimed at reducing intracellular GSH have been proposed in order to enhance the susceptibility of cancer cells to therapies [252].

The pioneer strategy is represented by buthionine sulfoximine (BSO), an irreversible inhibitor of GCL. Its use in combination with melphalan underwent Phase I clinical trials in neuroblastoma patients has been demonstrated to be well tolerated [253]. Moreover, it has been well described that BSO is effective in inducing cancer cell death [254,255,256,257] that recently has been recognised as ferroptosis [258]. However, the short half-life of BSO which would require prolonged infusions to keep constant BSO blood levels, and the occurrence of leukopenia and thrombocytopenia as side effects, may limit its clinical use [259]. Therefore, an alternative approach can be to reduce the availability of GSH precursors. A potential target is the glutamate/cystine antiporter system Xc− whose inhibition reduces the uptake of cystine and consequently GSH levels [235]. The Xc− inhibitor sulfasalazine was tested in a phase I/II study for the treatment of malignant gliomas but the study was early terminated due to lack of response and toxic effects [260].

Another Xc− inhibitor, erastin, was shown to enhance the pro-apoptotic action of tumour necrosis factor-related apoptosis-inducing ligand (TRAIL) in colon cancer cells [261]. Notably, also erastin activates ferroptosis, and it was hypothesised to contribute to the increased sensitivity of lung cancer cells to radiotherapy [262]. A phase I clinical trial testing the safety of erastin analogue PRLX 93,936 in patients with advanced solid tumours was completed in 2012 (NCT00528047). GSH precursors can be also obtained after degradation of extruded GSH by γGT. In this direction, γGT inhibitors such as glutamate analogs and boronate derivatives have been produced and tested, even if the problem of toxicity prevents their use in vivo [263].

### 5.3. Modulation of NRF2 Activity

#### 5.3.1. NRF2 Inducers

Most pharmacological NRF2 activators are electrophilic molecules and among these fumaric acid ester dimethyl fumarate (DMF) has been approved in 2013 by the United States Food and Drug Administration (FDA) for relapsing-remitting multiple sclerosis [264]. Additionally, two synthetic triterpenoids (CDDO-Im; 1- [2-cyano-3-, 12-dioxoleana-1,9 (11) -dien-28-oil] imidazole) and (CDDO-Me; bardoxolone methyl; methyl 2 -cyano-3,12-dioxooleana-1,9 (11) dien-28-oate) are strong inducers of NRF2 [265]. Although CDDO and its derivatives have been tested in several preclinical and clinical studies as anticancer and anti-inflammatory drugs, the high incidence of heart failure and the inability to identify their specific action mechanism have limited their extensive clinical use [266].

A recent work also highlights the potential use of different pharmacological NRF2 activators in the treatment of NASH [166].

Moreover, several compounds known as dietary phytochemicals are electrophilic NRF2 inducers. Curcumin, quercetin, genistein, epigallocatechin-3-gallate (EGCG), sulforaphane and resveratrol modify key cysteine residues on Keap1 protein by oxidation or alkylation, leading to NRF2 activation, and are characterised by few side effects [267]. Thus, by activating NRF2 and its target genes various oxidative-stress related diseases including type 2 diabetes, neurodegenerative diseases like AD and PD, CVD and kidney diseases can be mitigated/prevented by natural compounds [268].

Curcumin, extracted from *Curcuma longa* rhizomes, exerts strong anti-inflammatory, antioxidant and anti-viral effects [269]. In mice fed with high-fat diet curcumin attenuates glucose intolerance by activating NRF2 translocation and inducing HO-1 [270]. The activation of NRF2/HO-1 axis is also involved in the inhibition of NLRP3 inflammasome and the reduction of inflammation [271,272].

NRF2-dependent inhibition of NLRP3 is an anti-inflammatory mechanism shared by other compounds such as quercetin, sulforaphane and EGCG. Thus, quercetin [273], a flavonoid present in onions, apples, grapes, and berries exerts beneficial effects on lung, cardiovascular and neurodegenerative diseases by increasing NRF2 at mRNA level [268]. Sulforaphane, a well-known isothiocyanate abundant in cruciferous vegetables [266,274] has been proved efficacy in preventing angiotensin II-induced cardiomyopathy [275] and in improving diabetic retinopathy [276].

EGCG, the major catechin found in green tea, shows antioxidant and anti-inflammatory effects preventing lupus nephritis [277]. It exerts protection in rats with middle cerebral artery occlusion (MCAO) against cerebral ischemia-induced oxidative stress by activating NRF2 and its downstream genes HO-1, GCLC and GCLM [278].

Another natural compound able to activate NRF2 is resveratrol, a polyphenol present in berries, grapes, peanuts and red wine. Resveratrol has different biological proprieties including antioxidant, anti-inflammatory and antiplatelet aggregation potential [279]. It has been shown to protect oligodendroglial cells against LPS toxicity [280] and to confer neuroprotection against amyloid β1-42 [281] by modulating NRF2/HO-1 axis. Furthermore, it has been demonstrated that in streptozotocin-induced diabetic mice resveratrol was able to inhibit cardiac dysfunction and hypertrophy by the activation of NRF2 and the induction of its target genes, among which include HO-1 [282].

#### 5.3.2. NRF2 Inhibitors to Improve Therapy Susceptibility in Cancer

The inhibition of NRF2 could represent a strategy to improve cancer therapy. Among the inhibitors proposed so far unfortunately, some limitations have been highlighted. Brusatol has been widely studied for its ability to strongly reduce NRF2 protein levels in a Keap1-independent way [283]. Although the in vitro efficacy of brusatol has been widely proved, its use is limited since it has been shown to work as a global inhibitor of protein synthesis [284].

Trigonelline, another NRF2 inhibitor, is an alkaloid present in many food plants and seeds. It acts by blocking NRF2 phosphorylation (Ser40) and its nuclear import has been proved in non-small cell lung cancer cell lines [285]. The use of trigonelline is more promising but needs further investigation.

An alternative strategy could be to indirectly inhibit NRF2 by targeting the molecular pathways involved in its regulation. For instance, the use of PI3K inhibitors promotes NRF2 degradation via the GSK-3/β-TrCP axis as proved in Keap1-mutant cancer cell lines [286].

Moreover, a small molecule called ML385 specifically binds to the Neh1 domain of NRF2 interfering with its binding with sMAF. In NSCLC harbouring Keap1 mutation the use of ML385 in combination with platinum-based drugs substantially enhances the cytotoxicity increasing the therapeutic effects [287].

Among the limitations of NRF2 inhibitors in cancer therapy, it is important to highlight the potential toxicity exerted by NRF2 inhibitors on host healthy cell needs, and the possibility to hamper cancer immune recognition, as recently reviewed by De Nicola [288].

For this reason, the development of specific delivery systems to cancer cells could be a chance for targeting NRF2 pathway and achieving the best efficacy.

### 5.4. Vitamins

Recently, it has been surprisingly shown that a significant portion of the world’s population, even in developed countries, consume many vitamins and minerals (V/M) at levels below those recommended. Notably, modest V/M deficiencies, insufficient to elicit overt deficiency, might contribute significantly to the aging process and the diseases associated with aging [289].

Several studies and clinical trials have demonstrated that vitamins C and E have beneficial effects in reducing the risk of various oxidative stress-related diseases including cardiovascular diseases, cancer, diabetes, hypertension and cataracts [290]. Unfortunately, an almost equal number of studies showed no significant effect. While vitamin C and vitamin E are characterised by low toxicity and their administration at higher intake did not determine adverse effects, several in vivo animal studies showed that antioxidant supplements, including NAC, vitamin E and the soluble vitamin E analogue Trolox, promoted cancer development and metastasis, as above discussed and reported in [249].

Indeed, antioxidants could paradoxically enhance cancer cell survival and proliferation and/or inhibit the effects of anti-cancer treatments, which often act also through the induction of oxidative stress [291], and there is currently no consensus on the efficacy and safety of dietary antioxidant supplementation during conventional cancer therapy [292].

### 5.5. Dietary Supplementation with Antioxidants: Lights and Shadows of Their Effects

The evaluation of the efficacy of dietary antioxidant supplementation in the protection against oxidative stress related diseases is still largely inconclusive and sometimes even harmful effects of antioxidants have been revealed [293].

In the management of obesity and type 2 diabetes reasonable evidence suggests marginal benefits of supplementation with zinc, lipoic acid, carnitine, cinnamon, green tea, and possibly vitamin C plus E and antioxidant-rich foods are recommended as part of the lifestyle [294]. However, no real recommendation on the use of vitamin supplements in type 2 diabetes mellitus can be issued [295].

Supplementation with vitamins and antioxidants was not associated with reductions in the risk of major cardiovascular events, so there is no evidence to support their use [296]. An exception might be represented by lycopene in tomatoes. Epidemiological evidence suggests an association between consumption of tomato products or lycopene and lower risk for cardiovascular diseases. Overall, interventions supplementing tomato were associated with significant reductions in low-density lipoprotein (LDL)-cholesterol. These results support the development of promising individualised nutritional strategies involving tomatoes to tackle cardiovascular diseases [297]. Moreover, encouraging results have also been obtained with lycopene supplementation as chemoprevention of prostate cancer [298]. A recent review, analysing antioxidant supplementation in cardiovascular disease but also in all-cause mortality, seems to indicate that only when selenium was present in the antioxidant mixtures a reduced all-cause mortality was observed [299].

Furthermore, phytochemicals like curcumin, resveratrol, terpenoids, EGCG and isothiocyanates have been found to exert cytoprotective effects potentially beneficial in cardiovascular and neurological disorders via activation of the phase II detoxifying and antioxidant enzymes like HO- 1, GPX and GST by targeting NRF2 [300], as already discussed in Section 5.3.

In AD, although promising results have been found with antioxidant supplementation in animal models, the results in humans yielded equivocal results, even if epidemiological data linking diet and risk of AD is rapidly increasing [301].

In cystic fibrosis, antioxidant micronutrients supplementation does not appear to be a positive treatment on clinical end-points; however, glutathione supplementation showed some benefit to lung function and nutritional status [302].

Supplementation with several exogenous antioxidants to reduce oxidative stress and inflammation has been suggested in patients with renal replacement treatment. The data regarding the beneficial antioxidant effect of omega-3 fatty acids, statins, coenzyme Q10, curcumin, trace elements, vitamins B and D, green tea, flavonoids, and polyphenols remain controversial; NAC and α-tocopherol seem to have the most promising results in dialysis patients. However, currently none of these compounds are recommended [303].

Based on the collected findings, some factors can be taken into consideration in order to explain the failure of antioxidant supplementation, such as (i) patient genetic background, (ii) bioavailability of the molecules used, (iii) non-specific effects of antioxidants in human body and (iv) incorrect initial selection of the patients, mainly in elderly subjects.

However, although antioxidant therapy is characterised by “shadows” and “lights” that represent the negative and positive outcomes, some scientists conclude that it is possible to learn from failures, taking advantage of novel approaches such as mitochondria-targeting drugs and antioxidant gene therapy [304].

Moreover, other authors think that it might be crucial to determine the adequate intervention time of antioxidant supplementation and the proper antioxidant dosage dependent on the individual oxidative state [305].

## 6. Conclusions

The role played by ROS in pathophysiology appears very complex, and is strictly dependent on their levels, which are determined by cell antioxidant defence.

The modulation of antioxidants may have a double-edged impact on the different pathophysiological conditions (Figure 2):antioxidant supplementation is utilised to prevent the onset of many age-related diseases even though in most cases it does not appear effective and sometimes can produce harmful effects [293]. Notably, caloric restriction and intermitting fasting by increasing the threshold of stress adaptive resistance has been proposed as alternative strategy able to preserve the human body from diseases and to increase the life expectancy [306,307];antioxidant depletion can be employed in association with therapy in order to fight infections [308] or especially cancer [309,310]. In fact, as herein reported, several studies demonstrate that more malignant tumour cells display high levels of endogenous antioxidants [211,212] and become resistant to oxidative stress generated by long term-treatment with anticancer drugs [202,203,204,205,218]. Unfortunately, the sensitising therapy based on the reduction of the antioxidant defence could affect also healthy cells, leading to systemic side effects.

Considering the positive and negative aspects of antioxidant modulation, it could be crucial in the future to identify new intracellular redox-related targets and to monitor in real time the “right balance” of ROS in each patient under the different conditions of disease and of therapeutic treatment in order to improve strategies for the prevention and treatment of human diseases.

## Figures and Tables

**Figure 1 antioxidants-11-01613-f001:**
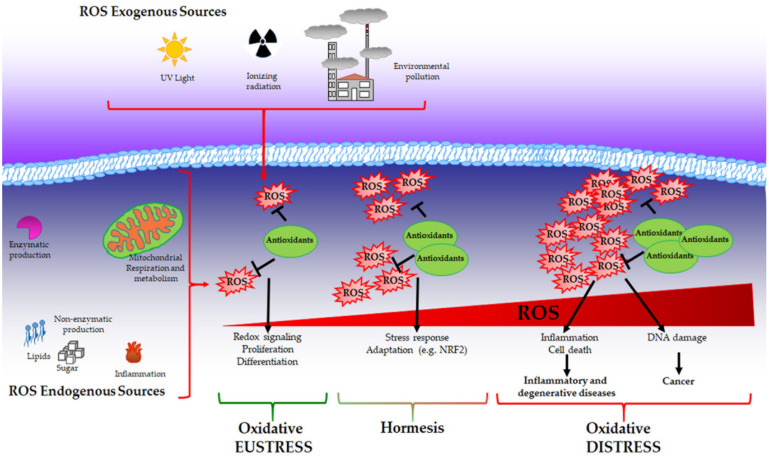
Oxidative eustress, hormesis and oxidative distress. Reactive oxygen species (ROS) can be produced by both exogenous and endogenous sources and are balanced by the presence of antioxidants that collaborate to maintain redox homeostasis. Under this condition called oxidative eustress, ROS act as pleiotropic signalling molecules playing a role in cell growth, proliferation and differentiation. When ROS production reaches moderate levels, a cell adaptive response is promoted to guarantee cell survival (hormesis). Supraphysiological ROS levels induce irreversible damage to biological macromolecules (oxidative distress), contributing to the onset of inflammatory, degenerative and neoplastic diseases.

**Figure 2 antioxidants-11-01613-f002:**
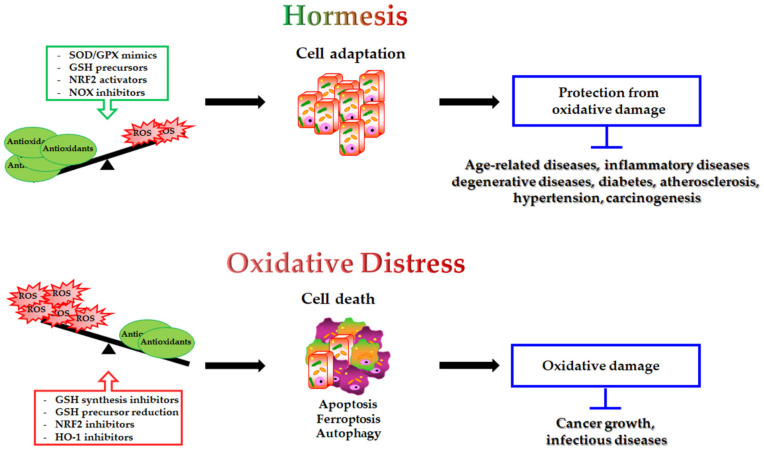
Double-edged role of antioxidant modulation and its impact on human health. On one hand, SOD/GPX mimics GSH precursors, NRF2 activators and NOX inhibitors by decreasing intracellular ROS levels, inducing cell adaptive response and protection from oxidative damage. This condition potentially prevents the onset of several human diseases. On the other hand, GSH synthesis inhibitors, reduced GSH precursor availability, NRF2 inhibitors and HO-1 inhibitors, by inducing oxidative distress, can be employed to trigger cell death (apoptosis, ferroptosis, autophagy) in the treatment of cancer and infectious diseases.

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
