# Peer review of "Hormesis and Oxidative Distress: Pathophysiology of Reactive Oxygen Species and the Open Question of Antioxidant Modulation and Supplementation"

_antioxidants, 2022, doi:10.3390/antiox11081613_

Round 1

Reviewer 1 Report

Reviewer comments

The authors deal with the existing problem between reactive oxygen species and the therapeutic and problematic potential of the use of antioxidant modulators and supplements. When reading the manuscript, sometimes the lack of a clear objective is observed and the ideas are mixed, without a clear line of thought. The structure of the manuscript should be revised to give a clearer line of reading. In addition, certain pathologies such as cancer and neurodegeneration are discussed in great detail, but others such as highly prevalent metabolic diseases are hardly mentioned. It happens in a similar way with the antioxidant mechanisms, treating glutathione in great detail but then going over other antioxidants. In this sense, if the authors expose a general review, they should balance the different sections treated or if they want to make it exhaustive, they should treat all the sections and components that are mentioned in a similar manner.

Some sentences with the main outcomes extracted from the analysis of the data available in the review should be included in the abstract.

Although hydrogen peroxide is not a radical species, it is a reactive oxygen species that, due to its half-life, has important implications as a cellular messenger. In this sense, it is important to treat this function in redox signalling in the introduction.

Line 92, all abbreviations should be described the first time used In this case, CYP450 should be defined.

Line 125, the sentence requires a reference.

When explaining the protection against free radical generation, the authors only discuss endogenous antioxidants and leave exogenous antioxidants undiscussed. In this sense, a brief subsection dealing with these antioxidants such as vitamins C and E, polyphenols, etc. should be included.

The role of non-coding RNAs or microRNAs in redox signaling is something new that should be discussed in more detail.

Line 340, a reference should be added.

Line 381, specific references should be added for each type of disorder indicated.

Lines 408-420, today the aging of society is a very relevant issue due to its socio-health implications. In this sense, this section should be worked on in more detail, exposing the controversies of Harman's aging theory.

When dealing with oxidative (di)stress and disease, the authors present some examples, but leave out others of great importance in today's society. In this sense, the high prevalence of obesity and its relationship with a prooxidative state should be addressed in the review.

At the beginning of the section antioxidants used for therapy some introductory sentences to better focus the aim should be added. Also, the information about SOD and GPX mimics, NOX inhibitors should be added as separated subsections and more developed explaining the mechanisms of action and characteristics.

Author Response

The authors deal with the existing problem between reactive oxygen species and the therapeutic and problematic potential of the use of antioxidant modulators and supplements. When reading the manuscript, sometimes the lack of a clear objective is observed and the ideas are mixed, without a clear line of thought. The structure of the manuscript should be revised to give a clearer line of reading. In addition, certain pathologies such as cancer and neurodegeneration are discussed in great detail, but others such as highly prevalent metabolic diseases are hardly mentioned. It happens in a similar way with the antioxidant mechanisms, treating glutathione in great detail but then going over other antioxidants. In this sense, if the authors expose a general review, they should balance the different sections treated or if they want to make it exhaustive, they should treat all the sections and components that are mentioned in a similar manner.

We thank the Reviewer for the suggestions that certainly will improve the manuscript.

  • Some sentences with the main outcomes extracted from the analysis of the data available in the review should be included in the abstract.

In agreement with the Reviewer, the text of the manuscript has been revised and also the abstract has been properly modified including the main outcomes of the treated topic.

2) Although hydrogen peroxide is not a radical species, it is a reactive oxygen species that, due to its half-life, has important implications as a cellular messenger. In this sense, it is important to treat this function in redox signalling in the introduction.

We agree with the Reviewer. In fact, we have already reported the function of hydrogen peroxide in redox signaling (see lines 165-205 of the revised manuscript).

3) Line 92, all abbreviations should be described the first time used In this case, CYP450 should be defined.

As required, all abbreviations have been checked and reported the first time used.

4) Line 125, the sentence requires a reference.

As required, a relevant reference has been added (ref. 11 in the revised manuscript)

5) When explaining the protection against free radical generation, the authors only discuss endogenous antioxidants and leave exogenous antioxidants undiscussed. In this sense, a brief subsection dealing with these antioxidants such as vitamins C and E, polyphenols, etc. should be included.

As suggested by the Reviewer, a brief subsection dealing with the exogenous antioxidants has been added in the Section 2 (lines 137-148).

6) The role of non-coding RNAs or microRNAs in redox signaling is something new that should be discussed in more detail.

As properly suggested by the Reviewer, a paragraph about the role played by miRNAs in redox signaling has been added in the section 3.3 (lines 261-293).

7) Line 340, a reference should be added.

As required, a relevant reference has been added (ref. 82 in the revised manuscript)

8) Line 381, specific references should be added for each type of disorder indicated.

The role played by NRF2 modulation in the different disorders is described in the section 3.2 in which specific references have been added (refs. 95-100 in the revised manuscript)

9) Lines 408-420, today the aging of society is a very relevant issue due to its socio-health implications. In this sense, this section should be worked on in more detail, exposing the controversies of Harman's aging theory.

We agree with the Reviewer and as suggested, the section concerning aging has been revised and the controversies of Harman's aging theory have been treated in more detail in the section 4.1 (lines 320-341).

10) When dealing with oxidative (di)stress and disease, the authors present some examples, but leave out others of great importance in today's society. In this sense, the high prevalence of obesity and its relationship with a prooxidative state should be addressed in the review.

In agreement with the Reviewer’s suggestion, a section 4.3 addressed to the relationship between obesity and pro-oxidative state has been added (lines 415-439).

11) At the beginning of the section antioxidants used for therapy some introductory sentences to better focus the aim should be added. Also, the information about SOD and GPX mimics, NOX inhibitors should be added as separated subsections and more developed explaining the mechanisms of action and characteristics.

As properly required, an introduction has been added at the beginning of the section 5 dedicated to the role of antioxidants in therapy in order to better focus the aim. Also, more informations about SOD and GPX mimics, and in particular of NOX inhibitors have been added in the section 5.1 (lines 566-590).

Reviewer 2 Report

I was very excited when the title came to me but disappointed after went through the manuscript. Generally speaking, this manuscript feels more like a book chapter rather than a review. The content lacks novelty. They authors tried to put too many eggs in one basket making the manuscript too long but failed to provide recent insights on a specific topic. Other detailed comments are as follows:

1. Free radicals and reactive oxygen species (ROS) are not equal. Section 1 is suggested to be concentrated on ROS considering the main topic of the manuscript. The title and subtitle of Section 2 should be revised.

2. Some definitions, such as oxidative eustress and oxidative distress, should be concisely introduced.

3. If there is only one point to be summarized or emphasized in a section, numerical order is not needed, e.g. Section 3 and Part 5, only 3.1, 5.2.1 and 5.4.1.

4. Section 4.2. The nuclear factor erythroid 2 p45-related factor 2 (NRF2): the authors described the research progress on NRF2 with 12 paragraphs. Summarization is strongly needed. Similar problems can be seen in other parts in the manuscript as well.

5. The quality of the two figures is not high enough.

6. Figure 2, Cell adaptation does not necessarily lead to increased life span; the title and figure captions are focused on human health and diseases, but only cancer was included in the figure.

7. Conclusions are not well focused on what they have viewed in the previous sections. In addition, the authors should be highly refined with their own insights and put up with future research prospects.

Author Response

I was very excited when the title came to me but disappointed after went through the manuscript. Generally speaking, this manuscript feels more like a book chapter rather than a review. The content lacks novelty. They authors tried to put too many eggs in one basket making the manuscript too long but failed to provide recent insights on a specific topic. Other detailed comments are as follows:

1) Free radicals and reactive oxygen species (ROS) are not equal. Section 1 is suggested to be concentrated on ROS considering the main topic of the manuscript. The title and subtitle of Section 2 should be revised.

In agreement with Reviewer’s suggestion, the section 1 is now focused on ROS. Moreover, the title of section 2 has been modified.

2) Some definitions, such as oxidative eustress and oxidative distress, should be concisely introduced.

Indeed, we have already defined the concept of oxidative eustress at lines 172-173 and that of oxidative distress at lines 201-205. Also in the legend of Figure 1, the definition of oxidative eustress, hormesis and oxidative distress has been reported. For more clarity, we have also added these definitions in the section 6 of the Conclusions.

3) If there is only one point to be summarized or emphasized in a section, numerical order is not needed, e.g. Section 3 and Part 5, only 3.1, 5.2.1 and 5.4.1.

In agreement with the Reviewer, sections and sub-sections have been checked and the numerical order has been deleted where it is not needed.

4) Section 4.2. The nuclear factor erythroid 2 p45-related factor 2 (NRF2): the authors described the research progress on NRF2 with 12 paragraphs. Summarization is strongly needed. Similar problems can be seen in other parts in the manuscript as well.

As suggested, the section 3.2. (ex section 4.2) of the revised version has been summarized.

5) The quality of the two figures is not high enough.

We are sorry that the reviewer did not find the quality of the images satisfactory. Indeed, following the instructions for authors, the submitted figure files are in TIF format and have a resolution of 300 dpi. However, we believe that the quality of the figures inserted in the text may be reduced as the consequence of the TIF image transfer within the body of the text. In order to overcome this drawback, both images have been now sent as TIF files.

6) Figure 2, Cell adaptation does not necessarily lead to increased life span; the title and figure captions are focused on human health and diseases, but only cancer was included in the figure.

In agreement with the Reviewer’s suggestion, the Figure 2 has been properly modified.

  1. Conclusions are not well focused on what they have viewed in the previous sections. In addition, the authors should be highly refined with their own insights and put up with future research prospects.

As properly suggested, we have better focused Conclusions based on our insights and future perspective.

Round 2

Reviewer 1 Report

In my opinion, the authors have adequately responded to my comments and the changes made to the manuscript are correct.

Author Response

We thank the Reviewer for having appreciated the revised manuscript.

Reviewer 2 Report

The revised manuscript has been improved. Some of my concerns have been properly addressed. However, I still feel the same as the first round of review. "This manuscript feels more like a book chapter rather than a review. The content lacks novelty. They authors tried to put too many eggs in one basket making the manuscript too long but failed to provide recent insights on a specific topic."

In addition, here are some minor points:

1.     As I have pointed out in the 1 round of review, if there is only one point to be summarized or emphasized in a section, numerical order is not needed. The authors revised what I have pointed out, but did not make a thorough revision. Please double check the manuscript.

2.     Some descriptions not accurate, e.g. line 450, only protein kinases were mentioned, but both protein kinases and transcription factors are modified as stated in the following sentence.

3.     5.3.1 Nrf2 inducer, CDDO compound is a strong and promising Nrf2 inducer, but not metioned.

4.     Double check English, e.g. sentence structure line 450-455.

Author Response

The revised manuscript has been improved. Some of my concerns have been properly addressed. However, I still feel the same as the first round of review. "This manuscript feels more like a book chapter rather than a review. The content lacks novelty. They authors tried to put too many eggs in one basket making the manuscript too long but failed to provide recent insights on a specific topic."

In this second phase of review, we have tried to emphasize the novelty of the content and in this direction we have further modified the abstract and rewritten the conclusions introducing a bulleted list.

We are fully aware that the topic per se is not the newest. However, it is certainly necessary to keep alive the attention on the role of antioxidants, because they are used both as preventing agents, with a little success, and as therapeutic ones, with harmful effects. In this regard, we are particularly interested because many of our preclinical studies (refs 197, 198. 205) have shown that chemoresistant tumor cells acquire a greater sensitivity to chemotherapy by lowering the endogenous antioxidant defense. Therefore, in the case of a patient affected by cancer, supplementation with exogenous antioxidants could have harmful effects capable of promoting tumor progression as reported in refs 291-293 and discussed in our studies (refs 211 and 212). On the other hand, as underlined in the conclusions of the revised version, this type of antioxidant-depleting approach could have detrimental effects on healthy cells with systemic damage.

Moreover, there is still a lot of work to do in order to better understand the above-reported issues. That is why it is important focusing again the attention on the pathophysiological role of ROS.

With this regard, as reported in the review’s conclusions, it could be important to establish in real time, for each patient, the right balance of ROS. We hypothesize that it might become a metabolic marker crucial to monitorate for guaranteeing good health and for improving the strategies of prevention or cure of several human diseases.

In addition, here are some minor points:

  1. As I have pointed out in the 1 round of review, if there is only one point to be summarized or emphasized in a section, numerical order is not needed. The authors revised what I have pointed out, but did not make a thorough revision. Please double check the manuscript.

As required, we have checked the manuscript and removed the unnecessary subsections.

  1. Some descriptions not accurate, e.g. line 450, only protein kinases were mentioned, but both protein kinases and transcription factors are modified as stated in the following sentence.

As required, the sentence has been properly modified (lines 202-207 of the new revised version)

  1. 3.1 Nrf2 inducer, CDDO compound is a strong and promising Nrf2 inducer, but not metioned.

As suggested, informations about the use of CDDO as NRF2 inducer has been added in the subsection 5-3-1 (lines 659-666 of the new revised version).

  1. Double check English, e.g. sentence structure line 450-455.

The English structure of the sentence reported at lines 450-455 has been checked and modified.
